

# A novel residual block: replace Conv1 × 1 with Conv3 × 3 and stack more convolutions

XiuJian Hu[1,*], Guanglei Sheng[1,2,*], Daohua Zhang[1] and Lin Li[1]

[1] Department of Electronic and Information Engineering, Bozhou University, Bozhou, Anhui, China
[2] School of Computer Science and Engineering, Xi'an University of Technology, Xi'an, Shaanxi, China
[*] These authors contributed equally to this work.

## ABSTRACT

The residual structure has an important influence on the design of the neural network model. The neural network model based on residual structure has excellent performance in computer vision tasks. However, the performance of classical residual networks is restricted by the size of receptive fields, channel information, spatial information and other factors. In this article, a novel residual structure is proposed. We modify the identity mapping and down-sampling block to get greater effective receptive field, and its excellent performance in channel information fusion and spatial feature extraction is verified by ablation studies. In order to further verify its feature extraction capability, a non-deep convolutional neural network (CNN) was designed and tested on Cifar10 and Cifar100 benchmark platforms using a naive training method. Our network model achieves better performance than other mainstream networks under the same training parameters, the accuracy we achieved is 3.08 percentage point higher than ResNet50 and 1.38 percentage points higher than ResNeXt50. Compared with SeResNet152, it is 0.29 percentage point higher in the case of 50 epochs less training.

## INTRODUCTION

In 2015, *He et al. (2016b)* from Microsoft Laboratory proposed a residual structure and designed ResNet deep neural network model, which won the first prize in the ILSVRC2015 (*Russakovsky et al., 2015*) Challenge for classification task and target detection, and the first prize in COCO (*Hansen et al., 2021*) target detection and image segmentation. The residual structure has a profound influence on the design of deep neural network models. Subsequently, many networks based on residual structure appeared, such as ResNetV2 (*He et al., 2016a*), WResNet (*Zagoruyko & Komodakis, 2016*), DenseNet (*Huang et al., 2017*), ResUnet++ (*Jha et al., 2019*), ConvNeXt (*Liu et al., 2022*). They have excellent performance in various downstream tasks in the field of computer vision, (*Ding et al., 2021*; *Jha et al., 2019*; *Jiang et al., 2018*; *Wang et al., 2017*; *Zhu & Wu, 2021*). *Wightman, Touvron & Jégou (2021)* once again demonstrated the advantages of residual structure through a large number of optimization experiments. The introduction of the Vision Transformer (VIT)

Corresponding author
XiuJian Hu,
2015020002@bzuu.edu.cn

(*Dosovitskiy et al., 2021*) soon replaced convolutional neural networks and became the most advanced image classification model. Staged Transformers, such as Swin Transformer *et al.* (*Ding et al., 2022*; *Liu et al., 2021*), reintroduced several priors of convolutional neural networks, which allow Transformers to serve as a general-purpose backbone effectively for CV tasks and demonstrate significant performance in a wide variety of visual tasks (*Cao et al., 2021*; *Chen et al., 2021*; *Valanarasu & Patel, 2022*). *Liu et al. (2022)* redesigned convolutional neural networks (CNNs) and tested the limits of what pure CNNs could achieve, which outperforms Transformer in terms of accuracy and scalability, proving that the performance of CNNs is still strong.

The feature extraction capability of CNNs is closely related to the size of convolutional kernels and affects the size of the receptive field. Large convolutional kernels will obtain larger receptive fields and be able to extract accurate high-level features (*He et al., 2016a*; *He et al., 2016b*; *Simonyan & Zisserman, 2015*), while small convolutional kernels have greater advantages in reducing network parameters and computational costs. The effectiveness of large convolution kernels has been verified (*Ding et al., 2022*; *Liu et al., 2022*; *Peng et al., 2017*; *Simonyan & Zisserman, 2015*) on some benchmark platforms, it is commonly believed that the size of convolution kernels should maintain a certain adaptation relationship with image size (*Ding et al., 2021*; *Ding et al., 2022*; *Han et al., 2021*; *Liu et al., 2022*). To obtain greater receptive fields and improve the robustness of the network (*Han et al., 2021*), the stack of three $Conv3 \times 3$ is designed as a new base residual structure to replace the combination of the stack of two $Conv3 \times 3$ and the combination of $Conv1 \times 1$ and $Conv3 \times 3$ in ResNet (*He et al., 2016a*; *He et al., 2016b*).

This article aims to verify the excellent performance of our new residual structures, our contributions are mainly about three aspects.

1. we propose a base residual structure and series of variants based on the stack of $Conv3 \times 3$ (Fig. 1), most of which put up an excellent performance in channel information fusion and spatial feature extraction.
2. we replace the $Conv1 \times 1$ to $Conv3 \times 3$ of the identity mapping, and the performance is significantly improved.
3. the performance of the non-deep CNNs based on our new residual structure is better than that of many deep networks, which provides valuable ideas for future network design.

## METHODS

In this section, we first introduce the latest research results for the choice of convolution kernel size. And then we discuss the issue of insufficient receptive field for the deep residual network, to do this we put forward a novel Base residual block based on the stack of $Conv3 \times 3$. Finally we propose an improved method of the Base residual structure.

### Kernel size

Neural network design needs to consider several key indicators of network depth, width, parameters, and computation (*Bau et al., 2020*; *Lorenz, 2021d*; *Xu, Duan & He, 2022*). Because the residual structure can solve the problems of network degradation well,

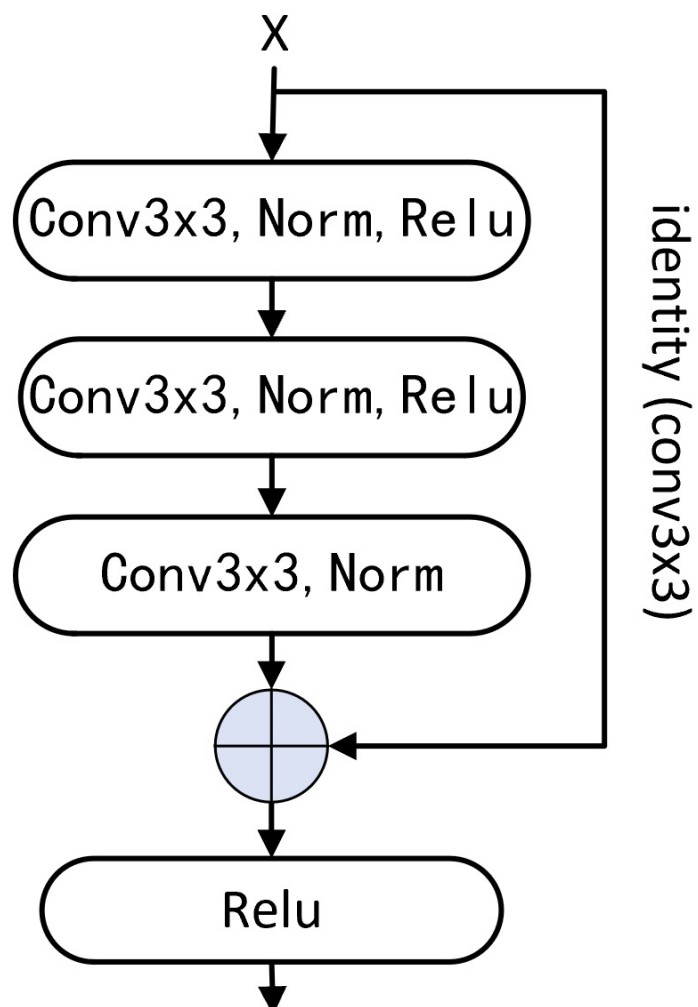

**Figure 1** **Basic residual block.** Using three Conv3 ×3 to replace two Conv3 ×3 in shallow residual net-work and the combination of Conv1 ×1 and Conv3 ×3 in deep network, and changing Conv1 ×1 to Conv3 ×3 of identity mapping.

increasing the network depth once became the main way to improve network performance for the lack of enough computing power (*Hu et al., 2020*; *Iandola et al., 2016*; *Tan, Pang & Le, 2020*; *Wu et al., 2021*). The design of the residual block is accomplished by skip connection with the identity mapping function. In ResNetV2 (*He et al., 2016a*), the author improved the residual structure and proved that the residual structure with the former standardized method has better performance without improving the effective receptive field. However, the application of the Transformer represented by the Vision Transformer (VIT) (*Dosovitskiy et al., 2021*) in the CV field shakes the dominant position of CNNs (*Ding et al., 2021*; *Ding et al., 2022*; *Liu et al., 2021*). Researchers begin to re-examine the performance space of CNNs and carry out comparative studies focused on the effective receptive field. *Ding et al. (2021)*; *Ding et al. (2022)*; *Han et al. (2021)*; *Liu et al. (2022)*

verified that using large convolutional kernels can significantly improve the performance of fully convolutional networks and achieve a state-of-the-art.

But what is the optimal convolution kernel size? The backbone network of RepLKNet is stacked with a large number of $ConvK \times K$ which can get a great enough receptive field, the max value of K as kernel size is 31 with corresponding to the image size of $1024 \times 2048$ (*Ding et al., 2021*). An interesting thing is that with the increasing size of the convolution kernel, the performance is still increasing by 1 to 2 points on Cityscapes (*Alexander, 2022*), ADE20K (*Zhou et al., 2017*) and other datasets while it's no longer improved on ImageNet (*Deng et al., 2009*). However, the image size of the ImageNet dataset is smaller than that of Cityscapes and ADE20K. The Similar phenomenon also exists in ConvNeXts and other full-convolutional neural networks with large convolutional kernels stacked (*Zhou, Jin & Zhang, 2014*; *Ding et al., 2021*; *Peng et al., 2017*). When those networks based on the large size of kernel work on datasets with small image size, they get worse performance (*Ding et al., 2022*; *Liu et al., 2022*). All of those give us reasons to believe that the selection of the convolution kernel size should keep match with the size of the image being processed in the specific task. Obtaining the appropriate receptive field is the primary factor in selecting the size of the convolutional kernel.

## Residual structure
### An introduction to the residual structure
The residual structure, also known as ResNet or residual network, is a neural network architecture introduced in 2015 to solve the issue of vanishing gradients. When the gradient signal is too small to be effectively propagated through the network during backpropagation, certain stages particularly those at the beginning of the network may become difficult to learn from (*He et al., 2016a*; *He et al., 2016b*). ResNets' key innovation is their "skip connections", which allow one or more stages in the network to be bypassed. This enables the network to learn residual mappings that add or subtract input to the output of a stage, rather than attempting to learn the entire mapping (*He et al., 2016b*). By using these skip connections, ResNets can train much deeper networks with greater accuracy, without being affected by vanishing gradients. ResNets have established state-of-the-art performance in various computer vision tasks, including image classification, object detection, and segmentation (*Wightman, Touvron & Jégou, 2021*).

### Bottleneck and identity mapping
The residual structure is the fundamental module of ResNet. In the ResNet network, the residual structure includes two parts: Bottleneck module and skip connection. The Bottleneck extracts feature information $F(x)$, and then adds it with skip connection $x$, and uses the activation function to nonlinearly activate $x + F(x)$. It can also be said that the identity mapping of feature information is achieved by using skip connections (*He et al., 2016a*). ResNet completes the forward propagation of information by stacking Bottleneck modules which consist of $Conv1 \times 1$ and $Conv3 \times 3$. Besides down-sampling, identity mapping is also important, *Han et al. (2021)* demonstrated that increasing the size of the convolution kernel without identity mapping resulted in a significant drop

in the performance (about 15% on ImageNet) while the identity mapping is added, the performance is improved.

### Kernel size and receptive field

There are many $Conv3 \times 3$ stacks in ResNet (*He et al., 2016b*); for example, ResNet-152 has 50 $Conv3 \times 3$ stages, while not getting an effective enough receptive field, which is because the residual block combined of $Conv1 \times 1$ and $Conv3 \times 3$ enables a down-sampling to be added to the identity map and activated after only once feature extraction. We believe that large convolution kernels can be replaced with several small convolution kernels, for example, one kernel with $7 \times 7$ size can be replaced with three $3 \times 3$ with more quickly inference speed, deeper network, and more advantages of nonlinear calculation (*Badrinarayanan, Kendall & Cipolla, 2017*; *Ding et al., 2021*; *Ding et al., 2019*; *Simonyan & Zisserman, 2015*).

## Improvement method

We have identified several areas for improvement, and have implemented the following measures to address them.

### Increase the receptive field size

The effectiveness of the convolution kernel with the size of $7 \times 7$ has been verified (*Han et al., 2021*; *Liu et al., 2022*), and the benefit of using a larger convolution kernel which plays a higher performance of the CNN is reflected in the latest research results. For example, the size of the convolution kernel is set to $31 \times 31$ (*Ding et al., 2022*). The residual structure designed by us based on the stack of three $Conv3 \times 3$ not only achieves the same size of receptive field as $Conv7 \times 7$, but also obtains a significant performance superior to $Conv7 \times 7$ in channel information fusion and feature extraction ability. To overcome the defects of the traditional deep network based on a small convolutional kernel, we redesigned a basic residual structure, using three $Conv3 \times 3$ to replace the combination of $Conv1 \times 1$ and $Conv3 \times 3$ in ResNet (*He et al., 2016a*; *He et al., 2016b*), beyond that we also change the identity mapping $Conv1 \times 1$ to $Conv3 \times 3$ (Fig. 1).

### Make the network lightweight

Since the effectiveness of depth-wise separable convolution (*Chollet, 2017*; *Howardet al., 2017*; *Sandler et al., 2018*) has been fully demonstrated in RepLKNet (*Han et al., 2021*), group convolution (*Ioannou et al., 2017*), asymmetric convolution (*Ding et al., 2019*) and point-wise convolution (*Howardet al., 2017*; *Sandler et al., 2018*) are used to design a series of variant structures excluding depth-wise separable convolution.

### The channel is squeezed and expanded

The performance of a convolutional neural network is mainly reflected in the feature extraction capability stage by stage (*Chen et al., 2018*; *Huang et al., 2017*; *Krizhevsky, Sutskever & Hinton, 2017*; *Russakovsky et al., 2015*). For a feature input (B, C, H, W), it is necessary to focus on the important features of the image, suppress unnecessary regional responses, and strengthen the attention of channel information and spatial information (*Dong, Cordonnier & Loukas, 2021*; *Guo et al., 2022*; *Loshchilov & Hutter, 2019*; *Vaswani*

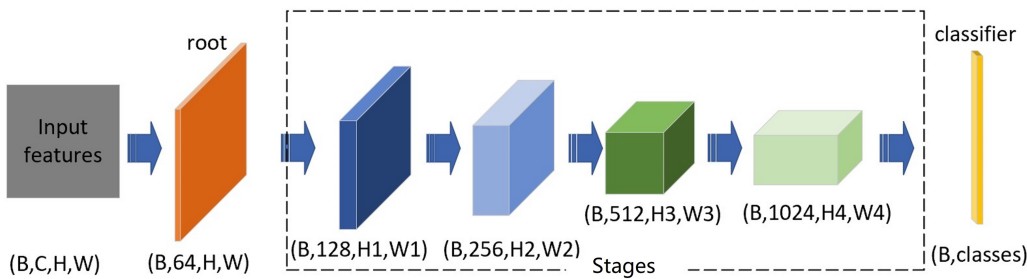

**Figure 2  Basic network structure.** The root block consists of just one Conv3 ×3, which converts to 64 channels; Layer1 converts the number of channels to 128 with optinal double down-sampling. Stage 2 to Stage 4 completes the double expansion of the number of channels and performs double down-sampling; the last block is classifier.

*et al., 2017*; *Woo et al., 2018*; *Wu et al., 2021*; *Zagoruyko & Komodakis, 2017*). To this end, we design the basic structure of a non-deep network and add attention mechanism and channel ES (expansion and squeeze) operation to the network structure (*Hu et al., 2020*).

### *Scale the depth and width of the network*
In terms of network depth and width selection, the design of a neural network tends to increase the depth to improve network performance while not significantly increasing computing costs. With the improvement of computing power, it is necessary and feasible to improve network width to a certain extent to keep the balance of compute cost and performance (*Lorenz, 2021c*; *Simonyan & Zisserman, 2015*; *Zagoruyko & Komodakis, 2016*).

## Structure design
To verify the performance of our novel residual block, a Base network is designed based on the residual network, and a series of variant structures are designed combining group convolution, channel information fusion, Channel Attention, and Spatial Attention mechanism (*Chen et al., 2020*; *Guo et al., 2022*; *Lin et al., 2022*; *Szegedy et al., 2017*; *Vaswani et al., 2017*; *Woo et al., 2018*).

## Basic network structure
We follow the design idea of residual structure and construct a non-deep network with large number of parameters with four stages. The excellent performance of the heavily parameterized network model has been verified on the benchmark platform of ImageNet (*Deng et al., 2009*) and other data sets, but few experimental results on the Cifar dataset. To keep the suitability of size between the image of Cifar datasets and the convolution kernel, we select the convolution kernel size as 3 × 3. Our basic network structure consists of a root module, four stages, and a classifier. The function of the root module is only to change the number of channels to 64 without double down-sampling; for the stages, each one consists of several residual blocks to enlarge the original number of channels by a factor of 2, the double down-sampling of the first stage is optional, and the next three stages' is fixed; the classifier module completes the output function (Fig. 2).

**Table 1 Basic network with five depth levels.** The max depth of our shallow network is 28 and the min is 13, the value of FLOPs is calculated by (128,3,32,32) input size and 100 classes.

| Level | Blocks | Convs | Parameter (M) | FLOPs(G) |
|---|---|---|---|---|
| Tiny_B | [1,1,1,1] | 12+1 | 108.03 | 2.52 |
| Tiny_L | [1,1,2,1] | 15+1 | 114.40 | 2.90 |
| Samll | [1,1,3,1] | 18+1 | 121.54 | 3.53 |
| Deep_B | [1,2,3,1] | 21+1 | 123.23 | 4.54 |
| Deep_L | [2,2,3,2] | 27+1 | 150.66 | 5.29 |

The width of each stage (including the root module) is the list of [64,128,256,512,1024]. The number of blocks, the parameters of the network, and the amount of calculation are shown in Table 1. We estimate the number of parameters by Torchsummary and the computation quantity of FLOPs by Thop. Since the root module includes one $Conv3 \times 3$, the number of convolution stages of each depth level is added by 1.

Due to our network's parameters being heavy, as the number of convolutional stages increases from 13 to 28, the FLOPs increase about twice, and that of parameters increases by about 42M. We combined asymmetric convolution (*Ding et al., 2019*), point-wise convolution, and group convolution (*Chollet, 2017*; *Ding et al., 2019*; *Lorenz, 2021b*; *Wang et al., 2017*) to design a series of variant structures to make the network lightweight (Fig. 3).

## Variant structure

A series of residual structure variants and network variants are designed from three aspects, namely network lightweight, channel information fusion (*Dumoulin & Visin, 2016*; *Howard & Ruder, 2018*; *Hu et al., 2020*; *Liu et al., 2022*) and attention mechanism (*Zhou, Jin & Zhang, 2014*; *Guo et al., 2022*; *Jaderberg et al., 2015*; *Vaswani et al., 2017*; *Woo et al., 2018*). All variants are designed on the condition of keeping the receptive field few changed. Six of them are residual structure variants (Fig. 3) and five are network variants. The variants are those we add an attention mechanism, and squeeze-expansion module between each stage of the basic network structure (*Cao et al., 2022*; *Hu et al., 2020*; *Liu et al., 2022*) (Fig. 2).

Based on the Base residual block, we change $Conv3 \times 3$ of the identity mapping to $Conv1 \times 1$ and named Variant1 (V1). Variant2 (V2) identity mapping is designed as $Conv1 \times 1$, and the second and third $Conv3 \times 3$ in the residual block are replaced with the combination of $Conv1 \times 1$ and $Conv3 \times 3$ (Group). Based on the Base residual block, V1 is modified to replace the third $Conv3 \times 3$ with asymmetrical convolution ($Conv1 \times 3$ and $Conv3 \times 1$) in the residual block named Variant3 (V3). Variant4 (V4) is modified based on V1 and we use the combination of $Conv3 \times 3$(Group) and $Conv1 \times 1$ to replace the third $Conv3 \times 3$ in the residual block; like the ConvNeXt's expansion module (*Liu et al., 2022*), Variant5 (V5) based on Base network is designed as adding a $4X$ channel expansion module between each stage. $Conv1 \times 1$ is used for the identity mapping of Variant6 (V6) and ES (expansion and squeeze) operations (*Hu et al., 2020*) are added in the residual block, and the channel is enlarged to an intermediate value ($mid = out + (out - in)/2$) by the first convolution operation, then the number of channels is reduced to OUT

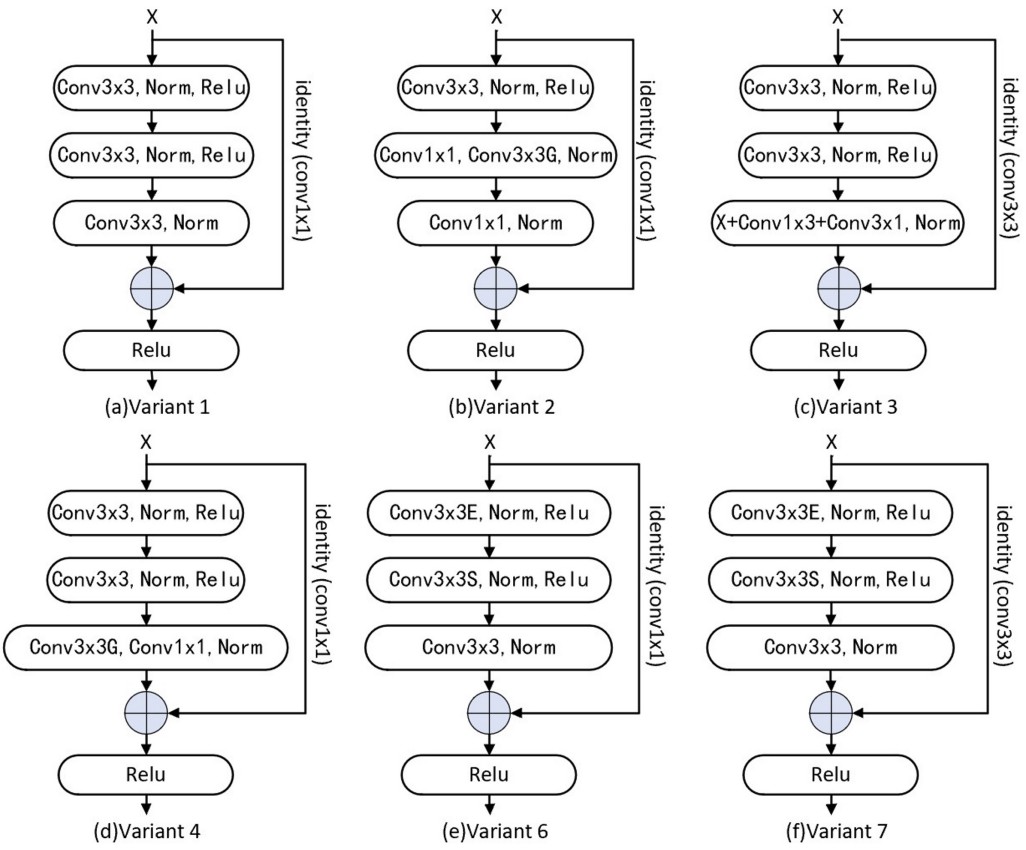

**Figure 3 Variant of residual block.** The combination of grouping convolution and point-wise convolution is used in V1, V2, V4 and V6 structures, and Conv3 $\times$3 in identity mapping is changed back to Conv1 $\times$1 to reduce the number of parameters. Asymmetric convolution is used to reduce the number of parameters in V3 structure. In V6 and V7, expansion channel is used to improve network performance.

by the second convolution operation. we change the identity mapping $Conv1 \times 1$ to $Conv3 \times 3$ based on Variant6 named Variant7 (V7). Based on Basic network, we add a Channel Attention mechanism between each stage named Variant8 (V8), and add CBAM (*ChannelAttention + SpatialAttention*) (*Woo et al., 2018*) to Base network between each stage named Variant9 (V9). Based on V1, we add a Channel Attention mechanism to it between each stage named V10. Unlike V10, the Spatial Attention mechanism (*Vaswani et al., 2017*; *Woo et al., 2018*) is added in Variant11 (V11).

Like the Base network, Each network variant is designed as five-level depth according to the number of residual blocks in stages (Table 2).

## EXPERIMENTS

This section covers three main aspects: (1) Experimental Setup, in which we introduce the selected dataset, validate its suitability, and explain the experimental parameters and their rationale. (2) Ablation Studies, where we design a series of experiments based on the 'Basic Model Structure' and 'Variant Structure' discussed in the 'Structure

**Table 2** Shows the number of parameters and FLOPs at the network's level of Tiny_B and Deep_B of the 10 network variants respectively, parameters are calculated in the same way as in Table 1 (Acc %).

| Alias | Variant | Level | Blocks | Parameter (MB) | FLOPs (G) |
|-------|---------|-------|--------|----------------|-----------|
| V1 | Skip_Conv1x1 | Tiny_B | [1,1,1,1] | 86.78 | 2.52 |
|    |              | Deep_B | [1,2,3,1] | 101.98 | 4.54 |
| V2 | Conv1x1_Conv3x3 (Group)_Conv1x1 | Tiny_B | [1,1,1,1] | 47.06 | 2.99 |
|    |              | Deep_B | [1,2,3,1] | 53.97 | 5.01 |
| V3 | Asymmetric | Tiny_B | [1,1,1,1] | 92.09 | 2.52 |
|    |              | Deep_B | [1,2,3,1] | 105.60 | 4.54 |
| V4 | Conv3x3(Group)_Conv1x1 | Tiny_B | [1,1,1,1] | 46.39 | 2.52 |
|    |              | Deep_B | [1,2,3,1] | 57.79 | 4.54 |
| V5 | Layer_SE | Tiny_B | [1,1,1,1] | 150.65 | 6.93 |
|    |              | Deep_B | [1,2,3,1] | 165.85 | 6.95 |
| V6 | Block_Expand | Tiny_B | [1,1,1,1] | 140.58 | 2.99 |
|    |              | Deep_B | [1,2,3,1] | 155.78 | 5.01 |
| V7 | Block_Expand_ Skip_Conv3x3 | Tiny_B | [1,1,1,1] | 161.83 | 2.99 |
|    |              | Deep_B | [1,2,3,1] | 177.03 | 5.01 |
| V8 | Layer_ChannelAtten | Tiny_B | [1,1,1,1] | 108.03 | 2.86 |
|    |              | Deep_B | [1,2,3,1] | 123.23 | 4.87 |
| V9 | Layer_CBAM | Tiny_B | [1,1,1,1] | 109.36 | 3.31 |
|    |              | Deep_B | [1,2,3,1] | 124.98 | 5.83 |
| V10 | Expand_ChannelAtten | Tiny_B | [1,1,1,1] | 140.58 | 3.33 |
|     |             | Deep_B | [1,2,3,1] | 155.78 | 5.34 |
| V11 | Expand_SpatialAtten | Tiny_B | [1,1,1,1] | 140.58 | 3.02 |
|     |             | Deep_B | [1,2,3,1] | 155.78 | 5.04 |

Design' section. We conduct experiments with various variant structures to demonstrate our improvement process and verify the effectiveness of our proposed measures. (3) Comparative Experiments, where we select several variants with different depths based on the results of the ablation studies and compare them with other mainstream networks to further validate the efficacy of our design method.

## Settings of the experiments
### Dataset selection

Our experiments aim to verify that the residual structure stacked by Conv 3×3 has an excellent performance in channel information fusion and spatial feature extraction because of getting suitable receptive fields, so we construct a series of residual structures and network variants in turn, and that all of those put up an excellent performance on Cifar10 and Cifar100 datasets compared with the current mainstream lightweight network, deep network, *etc*. The first reason for choosing the Cifar datasets is that each class of the Cifar100 dataset has 600 color images with a size of 32×32, among which 500 images are used as training sets and 100 images as test sets. Compared with ImageNet, its data volume and image size are much small. The second reason is that the relatively small picture size of Cifar datasets is more suitable for us to fully demonstrate the effectiveness and superiority of

our proposed method.. Therefore, we select the Cifar100 dataset as a benchmark platform and it comprehensively reflects the performance differences between different networks without any training skills and augmenting data.

### Training methods

The main performance of the network model depends on the design of the structure, the settings of training hyperparameters, and data augment skills (*He et al., 2019*; *Hoffer et al., 2019*; *Liu et al., 2022*; *Lorenz, 2021a*; *Zhang et al., 2018*). To fairly compare the performance of the model, we take into account two main factors: the task handled by the model and the benchmark platform. The same experimental environment and hyperparameter settings maybe not reflect the real differences in network model performance (*Ding et al., 2021*; *Ding et al., 2022*; *Strubell et al., 2017*), and so far there are no unified standards and methods to measure the fairness of comparison of a network model. To take the maximization of fair comparison and highlight the performance variation among the network architectures, we abandon any training skills across the experiment process, only use naive unified method for model training without any optimizing to the setting of hyperparameters, except for only one data augment method named RandomHorizontalFlip.

We conducted experiments on Python3.8.6, Pytorch1.8.2, and Cuda11.2 platforms, and trained the model on 2 Nvidia 3090 GPUs employing data-parallel computing. The benchmark test platform is Cifar10 and Cifar100 datasets with Epoch = 150, batchsize = 128, optimizer = SGD, momentum = 0.9, weight_decay = 5e−4, lr = 0.1, scheduler = MultiStepLR, milestones = [60,100,130], gamma = 0.1.

## Ablation study

According to the 'Basic model structure' and 'Variant structure' discussed in the Structure design section, 11 variants of network structure are designed relative to the Base network (Table 2), and six global random seeds are selected to test the stability of the Base network structure. Except for testing the stability of the network, others' random seeds are set to 1,234. According to the improved method we mentioned, the experiment includes the following aspects.

### Test for the stability of the Base network

We hope that the designed multiple variant structures can perform well in a robust network architecture, so we first test the robustness of the network architecture. Consistent with the above experimental settings, six global random seeds are selected to pass the test on the Cifar100 dataset (Table 3).

The above experimental results verify the robustness of our designed variant structure.

### Boost the receptive field size

To demonstrate that the Base structure design can boost the receptive field size, variant structures V1 and V2 are designed (Fig. 3), using Conv 1×1 in the identity map and Bottleneck, respectively. The experiment results are shown in Table 4.

Compared with Base, the number of parameters in V1 is reduced by 21.25M, and the accuracy is reduced by 1.41% on average. From the experimental comparison between

**Table 3  Results of stability tests (Acc %).** An average accuracy of 78.49% is obtained by training 150 epochs with 6 different random seeds, with a fluctuation of 0.3%.

| Seed | 1234 | 1008 | 1324 | 2413 | 4213 | 3412 |
|---|---|---|---|---|---|---|
| Max Accuracy | 78.38 | 78.20 | 78.36 | 78.45 | 78.66 | 78.86 |
| At Epoch | 138 | 119 | 113 | 113 | 139 | 118 |

**Table 4  Experimental results of group convolution, point-wise convolution and asymmetric convolution (Acc %).**

| Alias | Tiny_B | Tiny_L | Samll | Deep_B | Deep_L |
|---|---|---|---|---|---|
| Base | 78.38 | 78.88 | 78.52 | 79.03 | 79.03 |
| V1 | 77.59 | 76.72 | 77.45 | 77.65 | 77.39 |
| V2 | 75.85 | 76.81 | 77.75 | 77.96 | 77.32 |
| V3 $\checkmark$ | 78.05 | 79.09$^\uparrow$ | 79.41$^\uparrow$ | 79.34$^\uparrow$ | 79.28$^\uparrow$ |
| V4 $\checkmark$ | 77.37 | 78.37 | 77.99 | 77.47 | 77.57 |

the variant structure V1 and the base structure, it can be concluded that modifying the $Conv1 \times 1$ of the identity map in the residual structure to $Conv3 \times 3$ can significantly improve the performance of the network.

Based on V1, we construct V2 with the combination of $Conv1 \times 1$ and $Conv3 \times 3$ (Group) which improves the channel information fusion ability while reducing the receptive field of the bottleneck in the redisual structure. Compared with the Base, V2's accuracy is reduced by 1.63% on average, and the performance is reduced by 0.22% on average. From the experimental comparison between the variant structure V2 and the base structure, it can also be concluded that increasing the receptive field size of the bottleneck in the residual structure can significantly improve the performance of the network.

### Make the network lightweight

Although replacing Conv11 in the residual structure with Conv33 proved to be effective, the number of parameters of the network was high. To make the network more lightweight, we designed variant structures V3 and V4 (Fig. 3). (1) We construct V3 by the combination of $Conv3 \times 3$ (Group) and asymmetric ( $Conv1 \times 3 + Conv3 \times 1$) and verify the validity of asymmetric convolution, as the average number of V3's parameters is reduced about 17M, the average accuracy is improved by 0.3%, the max accuracy is improved by 0.89%. (2) V4 is modified from V1, the number of parameters is reduced to 50% of the Base, and the average accuracy decreases by 1.01% compared with it, but the average performance of V3 increases by 0.4%, and max increase by 0.65% compared with V1, it proves that the structure of V4 is effectiveness.

From the above analysis, it can be concluded that the performance of the identity mapping with $Conv3 \times 3$ is far superior to that of $Conv1 \times 1$. It is effective to replace the last $Conv3 \times 3$ with the combination of $Conv3 \times 3$ (Group) and asymmetric ($Conv1 \times 3 + Conv3 \times 1$), the combination of $Conv3 \times 3$ (Group) and $Conv1 \times 1$, we can reduce the net's parameter greatly, especially with the combination of the latter. By

**Table 5  Experiment results of feature fusion in channel (Acc %).**

| Alias | Tiny_B | Tiny_L | Samll | Deep_B | Deep_L |
|---|---|---|---|---|---|
| Base | 78.38 | 78.88 | 78.52 | 79.03 | 79.03 |
| V1 | 77.59 | 76.72 | 77.45 | 77.65 | 77.39 |
| V5 | 77.65 | 78.84 | 78.51 | 78.79 | 79.55$^\uparrow$ |
| V6 √ | 78.60$^\uparrow$ | 78.93 | 78.94$^\uparrow$ | 79.63$^\uparrow$ | 78.93 |
| V7 √ | 79.16$^{\uparrow\uparrow}$ | 79.03 | 79.50$^{\uparrow\uparrow}$ | 79.29$^{\uparrow\downarrow}$ | 79.53$^{\uparrow\uparrow}$ |

changing the identity mapping $Conv1 \times 1$ structure into $Conv3 \times 3$ in V4, we can improve its performance highly, so the structure design of V3 and V4 is retained.

### Enhance channel information fusion capability

In order to further improve the network performance, we make improvements in channel information fusion and design variant structures V6 and V7. The experiment results in this section are shown in Table 5. Referring to the practice of ConvNeXts (*Liu et al., 2022*), four times ES (4X expansion and squeeze) module was added between stages of the V5 structure. With the increase of network depth, the performance of V5 is gradually improved and surpassed at the fifth depth level (the accuracy was improved by 0.52% compared with Base). We adopts ES (expansion and squeeze) method in residual block to expand channel width (*Hu et al., 2020*; *Liu et al., 2022*; *Zagoruyko & Komodakis, 2016*) in the V6 structure. Compared with V1, the performance of V6 is improved by 1.57% on average and 2.21% on max, but the number of its parameters is increased by about 54M. Based on V6, the $Conv1 \times 1$ in identity mapping is changed to $Conv3 \times 3$ named V7, which improved by 0.3% on average and 0.95% on max compared with V6, and 0.534% on average and 0.98% on max compared with Base.

The above comparison results show that V5 has no advantages over V6 and V7 in a non-deep network, V5's preponderance perhaps needs to be verified in a deeper network. And that we can know from Table 5 that replacing $Conv1 \times 1$ with $Conv3 \times 3$ in the identity mapping can significantly improve network performance.

From the above analysis, it can be concluded that ES can improve the network performance, so we keep the design scheme of variant structure V6 and V7.

### Comparison with attention mechanism

In order to further verify the effectiveness of ES, some attention mechanisms are designed into the network architecture and compared with ES. The experiment results in this section are shown in Table 6. Spatial Attention and Channel Attention are added to the Base network to observe the performance of the network. It is considered empirically that adding an attention mechanism (*Celebi, Kingravi & Vela, 2013*; *Guo et al., 2022*; *Vaswani et al., 2017*) module is effective. V8 and V9 based on the Base network are added Channel Attention mechanism and CBAM(*Woo et al., 2018*) between stages respectively while it is lower compared with the Base's performance. Based on V6, the Channel Attention and the Spatial Attention mechanism (*Dong, Cordonnier & Loukas, 2021*; *Guo et al., 2022*; *Vaswani et al., 2017*; *Wojna et al., 2017*; *Woo et al., 2018*; *Xie et al., 2017*) are added to V10 and V11

**Table 6 Experimental results of Attentional Mechanism and ES (Expansion and Squeeze) modules based on the Base and V6(Acc %).**

| Alias | Tiny_B | Tiny_L | Samll | Deep_B | Deep_L |
|---|---|---|---|---|---|
| Base | 78.38 | 78.88 | 78.52 | 79.03 | 79.03 |
| V6 | 78.60 | 78.93 | 78.94 | 79.63 | 78.93 |
| V8 | 75.85 | 76.81 | 77.75 | 78.87 | 78.73 |
| V9 | 78.18 | 78.51 | 78.42 | 78.80 | 78.85 |
| V10 | 77.82 | 78.10 | 78.17 | 78.44 | – |
| V11 | 78.52$^{\uparrow\downarrow}$ | 79.28$^{\uparrow\downarrow}$ | 78.75$^{\uparrow\downarrow}$ | 79.32$^{\uparrow\downarrow}$ | 79.18$^{\uparrow\uparrow}$ |

between the stages their own, respectively. V10 is lower compared with Base and V6, and V11 Added Spatial Attention is slightly higher than Base's performance but is also lower than V6's.

Through the experimental comparison, we can see that adding various attention mechanisms and ES (expansion and squeeze) operations between stages can get no improvement in the performance of the Base and V6 residual structure. So the conclusion can be drawn that the residual structure based on the stack of three $Conv3 \times 3$ is powerful enough in channel fusion ability and spatial feature extraction ability (*Dong, Cordonnier & Loukas, 2021*) at current depth.

Over all, we can draw some conclusions as follows. (1) The residual structure based on the stack of $Conv3 \times 3$ has excellent performance. (2) It does not need extra attention mechanisms in channel information fusion and spatial feature extraction (*Dong, Cordonnier & Loukas, 2021*), but ES is necessary. (3) We can lightweight the network and achieve excellent performance without reducing the receptive field. (4) Replacing $Conv1 \times 1$ to $Conv3 \times 3$ in identity mapping results in performance improvements in multiple variants. Therefore, we retain five residual structures, including Base, V3, V4, V6 and V7, to provide a basis for optimizing networks with higher performance.

## Comparative experiments

In order to show the performance of our designed module, we selected our Base, V3, V4, V6, V7 and Vgg19bn, ResNet18 ResNet50, ResNeXt50, GoogLeNet, MobileNetv2, SeResNet50, ShuffleNetv2, *et al.* to make a full comparison with the experimental results under the same and different hyperparameter setting and training method. It is verified that the residual structures based on the stack of multiple $Conv3 \times 3$ put up excellent performance (*Fang et al., 2020*; *Xie et al., 2017*; *Zhu & Wu, 2021*). The intuitive results of the residual block on Cifar datasets we designed in contrast to other network can be seen in Fig. 4.

Under the same training hyperparameter condition, on the Cifar10 dataset, the performance of our simple non-deep network is comprehensively superior to Vgg19BN, ResNet18, ResNet50, ResNeXt50, GoogLeNet, MobileNetv2, SeResNet50, ShuffleNetv2- et al.. The lightweight models V3 and V4 achieve similar accuracy to ResNet18V2, but V3_Tiny-B and V3_Deep_B on Cifar100 dataset are 0.74% and 2.04% higher than ResNet18 respectively (Table 7).

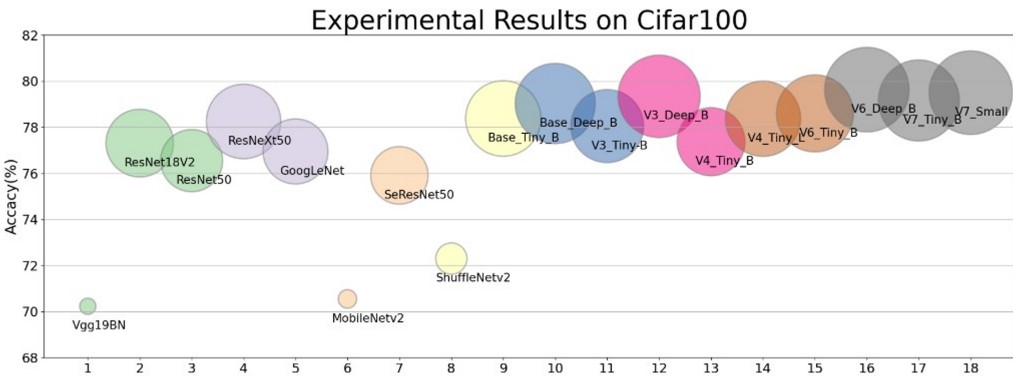

**Figure 4** **The performance of the shallow networks based on the residual block we designed contrast to the mainstream network.** Without any training skills, under the same experimental conditions and with the same parameter settings, the performance of the shallow network we designed exceeds that of the mainstream network (see Table 7).

Comparing the results of 150 epoch-trained experiments on the Cifar100 dataset, V6_Deep_B exceeds ResNet50 by a maximum of 3.08%, ResNeXt50 by a maximum of 1.38%, and SeResNet152 by a maximum of 0.29%. This is achieved without massively increasing the network depth, and the network structure can be further optimized to improve the performance.

Through the above analysis, we find that the network architecture based on the improved residual structure is competitive with the mainstream network architecture.

## CONCLUSIONS AND FUTURE WORK

We believe that obtaining the appropriate receptive field is the primary factor in selecting the size of the convolutional kernel. Our work mainly includes the following aspects. (1) A base residual structure and series of variants based on the stack of Conv 3×3 were proposed. (2) The method of replacing the Conv 1×1 with Conv 3×3 of the identity mapping can improve the performanceof residual structure. (3) The performance of the non-deep CNN based on our new residual structure is better than that of many deep networks, which provides valuable ideas for future network design. It was verified by experiments that the residual structure stacked by Conv 3×3 has an excellent performance in channel information fusion and spatial feature extraction because of getting suitable receptive fields, so we constructed a series of residual structures and network variants in turn, and that all of those put up an excellent performance on the Cifar10 and Cifar100 datasets compared with the current mainstream lightweight network, deep network, *etc.*

To keep the suitability of size between the receptive field and image size in the designing of a network, we stacked three $Conv\,3 \times 3$ to construct a base residual structure that can enhance the receptive field while maintaining strong channel information fusion capability.

**Table 7 Comparison of results on the benchmark platform (Acc %).** The initial $lr = 0.1$, scheduler = MultiStepLR, gamma=0.1, $batchsize = 128$, $weight\_decay = 5e-4$, $momuentu = 0.9$. All parameter counts are calculated with [28,3,32,32] input data size.

| Model | Convolutions | Parameter (MB) | Cifar10 150 epoch | Cifar100 | |
|---|---|---|---|---|---|
| | | | | 150 epoch | 200 epoch |
| Vgg19BN | 19 | 150.02 | 93.03 | 70.23 | 72.23 |
| ResNet18(V1,V2) | 18 | 42.80 | 94.86(V2) | 77.31(V2) | 75.61(V1) |
| ResNet50 | 50 | 90.43 | 94.39 | 76.55 | 77.39 |
| ResNeXt50 | 50 | 56.41 | 94.66 | 78.25 | 77.77 |
| ResNet152 | 152 | 147.33 | – | – | 77.69 |
| GoogLeNet | 22 | 24.42 | 94.54 | 76.95 | 78.03 |
| MobileNetv2 | 88 | 9.04 | 91.58 | 70.55 | 68.08 |
| SeResNet50 | 50 | 100.18 | 94.31 | 75.91 | 77.93 |
| SeResNet152 | 152 | 248.13 | – | – | 79.34 |
| ShuffleNetv2 | – | 5.19 | 92.10 | 72.30 | 69.51 |
| Base_Tiny_B (ours) | 13 | 108.03 | 95.25$^\uparrow$ | 78.38$^\uparrow$ | 78.63$^\uparrow$ |
| Base_Deep_B (ours) | 22 | 123.23 | – | 79.03$^\uparrow$ | 79.26$^\uparrow$ |
| V3_Tiny-B (ours) | 20 | 92.09 | 94.96$^\uparrow$ | 78.05$^\uparrow$ | 78.67$^\uparrow$ |
| V3_Deep_B (ours) | 35 | 105.60 | – | 79.34$^{\uparrow\uparrow}$ | 79.76$^\uparrow$ |
| V4_Tiny_B (ours) | 17 | 46.39 | 94.79$^\uparrow$ | 77.37 | 78.24 |
| V4_Tiny_L (ours) | 21 | 51.42 | – | 78.37$^\uparrow$ | |
| V6_Tiny_B (ours) | 13 | 140.58 | 95.16$^\uparrow$ | 78.60$^\uparrow$ | – |
| V6_Deep_B (ours) | 22 | 155.78 | – | 79.63$^{\uparrow\uparrow}$ | – |
| V7_Tiny_B (ours) | 13 | 161.83 | 95.23$^\uparrow$ | 79.16$^\uparrow$ | – |
| V7_Small (ours) | 19 | 175.34 | – | 79.50$^{\uparrow\uparrow}$ | – |

We will further study the adaptive relationship between the receptive field size and the image size to find a method to automatically adjust the convolution kernel size.

### Funding

This work was supported by the Natural Science Foundation of Colleges and Universities of Anhui Province (No. KJ2020A0773) and the Excellent top-of-the-line Talent Training Program of Colleges and Universities of Anhui Province (No. gxgnfx2019063). The funders had no role in study design, data collection and analysis, decision to publish, or preparation of the manuscript.

### Grant Disclosures

The following grant information was disclosed by the authors:
Natural Science Foundation of Colleges and Universities of Anhui Province: KJ2020A0773.
Excellent top-of-the-line Talent Training Program of Colleges.
Universities of Anhui Province:  gxgnfx2019063.

## Competing Interests

The authors declare there are no competing interests.

## Author Contributions

- XiuJian Hu conceived and designed the experiments, performed the computation work, prepared figures and/or tables, authored or reviewed drafts of the article, and approved the final draft.
- Guanglei Sheng conceived and designed the experiments, prepared figures and/or tables, authored or reviewed drafts of the article, and approved the final draft.
- Daohua Zhang performed the experiments, prepared figures and/or tables, authored or reviewed drafts of the article, and approved the final draft.
- Lin Li analyzed the data, prepared figures and/or tables, authored or reviewed drafts of the article, and approved the final draft.

## Data Availability

The code is available in the Supplemental Files.

The Cifar10&100 dataset is available on Figshare: Hu, Xiujian (2022): cifar10&cifar100. figshare. Dataset. https://doi.org/10.6084/m9.figshare.21532920.v2.

The original datasets are available at the University of Toronto: https://www.cs.toronto.edu/~kriz/cifar.html.

## Supplemental Information

Supplemental information for this article can be found online at http://dx.doi.org/10.7717/peerj-cs.1302#supplemental-information.

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
