# Peer review of "A novel residual block: replace Conv1× 1 with Conv3×3 and stack more convolutions"

_PeerJ Computer Science, doi:10.7717/peerj-cs.1302_

## Round 0.1 · original submission · Major Revisions

Three reviewers gave comments and suggested revisions. Therefore, please revise your manuscript and submit a revised version as soon as possible.

Reviewer 1 ·

Basic reporting

1. The number of network layers and the size of blocks optimized in this work are not innovative enough. It would be better if there were some innovations in the network structure.
3. Check grammar errors on line 25 and 28 in your abstract.
4. The authors should summarize the main contributions as well as the weaknesses and strengths of the study in the Conclusion section.
5. The writing of the paper needs improvement.

Experimental design

2. Conduct experiments on more datasets to verify the generalization of the proposed method

Validity of the findings

no

Additional comments

no

Reviewer 2 ·

Basic reporting

no comment

Experimental design

no comment

Validity of the findings

no comment

Additional comments

I would be very glad to re-review the paper in greater depth once it has been edited because the subject is interesting. However, the following problems need to be modified:
1.I would be very glad to re-review the paper in greater depth once it has been edited because the subject is interesting.
2.Try to set the problem discussed in this paper in more clear, write one section to define the problem。
3.The experiment is the major problem in the paper. The dataset is published, but the description is very rough.
4.The list of references is not in our style, please refer to a pdf file with "PeerJ-research-manuscript-template" which shows examples.
5. The figures and tables in this article need to be more consistent with our style.
6. There are less English grammar problems in the text. English grammar, spelling and sentence structure should be further improved so that the objectives and results of the research are clearly understood by the reader.
7. If the algorithm can be compared with the previous work in more aspects, it will be better to judge the improvement effect of the algorithm on the present work.
8. It is recommended that the content of the variant structure design section be described more clearly.
9. If the application in more downstream tasks can be supplemented, the performance of the algorithm will be better demonstrated.

Reviewer 3 ·

Basic reporting

This is a well-written paper containing interesting results which merit publication. For the benefit of the reader, however, a number of points need clarifying and certain statements require further justification. There are given below.
1. The authors should put forward more theoretical explanations for the results of his experiments.
2.The author should concentrated on the new algorithm with your idea and explained its advantages clearly with a most simple words.
3. The working platform of the project is too little introduced in this paper, so relevant descriptions need to be enriched.
4. The style of the reference in this paper is different from ours, please refer to our standard for modification.
5. I think it is necessary to introduce relevant theoretical basis in ablation experiments, which will make the paper easier to be understood and accepted by readers.
6. In this paper, there are 11 network structures and variations of residual structures. It is suggested to simplify the number of variant structures or to describe them more systematically.
7. There are a few grammar problems with sentence structure, verb tense, and clause construction. I suggest that the missing grammar in the text needs to be further improved.
8 Finally, I think the main contribution needs to be described more succinctly.

Experimental design

Experiments are well-designed.

Validity of the findings

The findings are interesting.

Additional comments

No additional comments.

---

## Round 0.2 · Major Revisions

Please see the comments from reviewer 1, which ask for a set of changes.

In addition:

The grammar in the paper needs work, as this is not idiomatic English, starting in the abstract and continuing throughout the paper. For example, there should always be a space before an open parenthesis, such as used in references.

At least the start of the paper needs to be more clearly written to appeal to a general computer science audience. The paper should explain:
* What is the residual structure?
* How is it related to a neural network model?
* How are convolutional kernels related to neural networks and residuals?
* Where do stacks fit into this process?
Some of this is shown in the figures, but it also should be explained in the text.

Regarding the experiments, please explain the logic behind the experiments that were run. Why were these experiments chosen? Please explain what is learned, overall, from each, in a qualitative sense rather than just providing quantitative results, as well as what is learned from the entire set, again qualitatively.

Reviewer 1 ·

Basic reporting

The experimental details should not be discussed in the conclusion chapter, but a high level summary of the problem should be given.

Experimental design

The selection of datasets should not be oriented by your method. Try to ensure the dataset you used can fully demonstrate the effectiveness and superiority of your proposed method.

Validity of the findings

Verify the effect on the real-world dataset of the actual application

Annotated reviews are not available for download in order to protect the identity of reviewers who chose to remain anonymous.

Reviewer 2 ·

Basic reporting

Professional English used throughout. Professional article structure, figures, tables.

Experimental design

Research question well defined, relevant & meaningful. Methods described with sufficient detail & information to replicate.

Validity of the findings

All underlying data have been provided; they are robust, statistically sound, & controlled.

Reviewer 3 ·

Basic reporting

My quesions are all addressed.

Experimental design

Well-designed.

Validity of the findings

The findings are interesting.

Additional comments

No additional comments.

---

## Round 0.3 · accepted · Accept

The paper is much improved.